# Severe Case of Delayed Replantation of Avulsed Permanent Central Incisor: A Case Report with Four-Year Follow-Up

**DOI:** 10.3390/medicina56100503

**Published:** 2020-09-25

**Authors:** Luísa Bandeira Lopes, João Botelho, Vanessa Machado

**Affiliations:** 1Pediatric Department, Centro de Investigação Interdisciplinar Egas Moniz, Egas Moniz Cooperativa de Ensino Superior, 2829-511 Almada, Portugal; 2Evidence-Based Hub, Clinical Research Unit (CRU), Centro de Investigação Interdisciplinar Egas Moniz (CiiEM), Instituto Universitário Egas Moniz (IUEM), 2829-511 Almada, Portugal; jbotelho@egasmoniz.edu.pt (J.B.); vmachado@egasmoniz.edu.pt (V.M.); 3Periodontology Department, Clinical Research Unit (CRU), Centro de Investigação Interdisciplinar Egas Moniz (CiiEM), Instituto Universitário Egas Moniz (IUEM), 2829-511 Almada, Portugal

**Keywords:** avulsion, replantation, dental traumatology, dental trauma, periodontal tissues, oral health

## Abstract

Avulsion is one of the most serious dental traumatic injuries with a reserved prognosis. This case report describes multiple trauma lesions in permanent central incisors of an eight-year-old girl and a four-year follow-up. The right upper incisor suffered avulsion, remained 16 h extraorally, and was replanted after extraoral endodontic therapy. The left maxillary central incisor suffered a noncomplicated crown fracture with concomitant subluxation. The present case adds to the literature a rare occurrence of success in a severe case with poor prognosis. For this reason, the International Association for Dental Traumatology (IADT) guidelines should be followed and, even in extreme situations, replantation should always be considered.

## 1. Introduction

Tooth avulsion is a complete loss of attachment between the tooth and the socket with complete rupture of the periodontal ligament and cemental damage [1,2]. The periodontal ligament cells’ vitality is the most important factor to allow the reattach following tooth replantation [1]. Furthermore, other factors like extraoral time, storage of the avulsed tooth, and maturity of the root are also important to dental prognosis [3,4,5,6,7,8,9,10].

Trauma can affect several tissues such as periodontal ligament, cementum, alveolar bone, and pulp. Additionally, some healing complications are strongly related with the presence of bacteria around the traumatized tooth. In this sense, antibiotics might be indicated due to their moderate effectiveness in the control of infection [11].

In severe avulsion cases, the periodontal ligament suffers dehydration, and if it lasts more than 60 min is unlikely to heal [2,12]. Therefore, immediate replantation of avulsed permanent teeth is currently the gold standard action measure [2,12]. However, replantation presents potential complications, namely, pulp necrosis, infection-related (inflammatory) resorption, and ankylosis-related (replacement) resorption [6,7,8,13,14,15,16].

In this paper, we report a case of multiple dental trauma on upper central incisors in an eight-year-old girl. Considering its uniqueness, this report will focus on the replantation of the right upper central incisor that suffered avulsion and remained 16 h out of mouth. Additionally, we present a clinical follow-up of four years without any common after-effects of avulsion. We want to highlight the importance of the International Association for Dental Traumatology (IADT) Guidelines, in order to obtain the best possible prognosis.

## 2. Case Presentation

Informed consent was obtained from her parents so that case records could be made available for teaching purposes, including scientific publication. All procedures followed the Helsinki Declaration, as reviewed in 2013, approved by the Egas Moniz Ethics Committee (Process 742, on April 2019).

An eight-year-old girl was attended to by the Pediatric Department at Egas Moniz Dental Clinic (Almada, Portugal), for an emergency appointment after multiple orofacial trauma. The patient presented no relevant medical history. Regarding the trauma event, her mother reported it having occurred 16 h ago, “where the tooth popped out of the mouth” (sic). After being identified as upper right central incisor (1.1), it stayed six hours in dry conditions and the remaining 10 h laid in milk. Intraoral and radiographic examination detected avulsion of the upper right central incisor (1.1) and uncomplicated crown fracture of the upper left central incisor (2.1) (Figure 1). During thermal tests, 2.1 showed exacerbated response to percussion, and no response to thermal. Periapical X-ray revealed complete avulsion of 1.1 and incomplete root development of 2.1 as well the absence of periapical lesions. Considering the latter, the diagnosis was the avulsion of tooth 1.1 and noncomplicated crown fracture concomitant with subluxation of tooth 2.1.

After determining the diagnosis, delayed replantation of 1.1 was suggested. This option was suggested given the possible functional and esthetic consequences to the patient with this tooth missing. Additionally, with the tooth present, bone volume would be preserved for future rehabilitation with dental implants. For this reason, the following steps were performed: (1) extraoral pulpectomy; (2) replantation; (3) splint with flexible orthodontic wire and composite.

Firstly, tooth 1.1 was cleaned and wiped with gauze. Next, root canal treatment was performed extraorally prior to replantation, with apexification with Mineral Trioxide Aggregate (MTA), obturation, and final restoration (Figure 2). Root canal was cleaned with 5.25% sodium hypochlorite (NaOCl) followed by 17% liquid ethylenediaminetetraacetic acid (EDTA). The canal was then dried, MTA was placed for apexification, and it was filled with gutta percha using a cordless obturation device for optimal backfill (SuperEndo-β, B&L Biotech). Then, gutta percha was covered with Vitrebond-Plus (3M ESPE, Maplewood, MN, USA) and the tooth restored with composite Filtek Z250 (3M ESPE, Maplewood, MN, USA).

Before replantation, the tooth was kept in a 2% sodium fluoride (NaF) solution for 20 min, because of the possibility to slow down osseous replacement [9,12]. Simultaneously, local anesthesia without vasoconstrictor (3% Mepivacaine) was delivered and the socket irrigated with saline solution. After confirming the absence of bone fractures inside the alveolus, 1.1 was gently replanted, radiographically confirmed, and then splinted with a flexible orthodontic wire and flowable composite. Amoxicillin was prescribed (every 12 h, for 8 days), plus mouthrinse with chlorhexidine gluconate (0.2%). We also confirmed if the tetanus vaccine was up to date. Soft diet was advised for 2 weeks and a careful brushing with a soft toothbrush.

At two weeks of follow-up, the splint was removed and pulp status remained unchanged in both teeth. At three months of follow-up, we performed clinical, radiograph, and sensibility tests. No resorption or infection was observed in 1.1. Pulp necrosis was diagnosed in pulp status of 2.1, and apexification with MTA was carried out in a single appointment.

Every six months, clinical and radiographic exams were performed. At four-year follow-up, we observed minor ankylosis-related (replacement) root resorption at the apical zone of the avulsed tooth (Figure 3) and some aesthetic concerns caused by discoloration due to MTA [17,18]. Resorption, infection, response to percussion, or infra-occlusion were not detected. To overcome the esthetic issues, two composite veneers were proposed considering the facial cranial growth stage.

## 3. Discussion

This case stresses the importance of the potential of replantation procedures, even in severe borderline cases such as this one. Even though, dental traumas are clinically unpredictable and a clinical challenge [19].

Successful replantation demands a rapid and strict fulfilled criteria to minimize periodontal and pulpal impact [6]. Nevertheless, the success of this case goes against all expectations given the time the tooth remained dried and out of mouth [20]. The decision for an extraoral endodontic treatment was based on the lower risk of external infection-related (inflammatory) root resorption and compliance of the patient [9,21]. Splinting was key to maintaining the tooth in a correct position and protects pulpal and periodontal tissues when the tooth suffers soft mobility and function [2,12]. Regarding the possible complications of a delayed replantation, only ankylosis-related (replacement) resorption was detected, making this case a fairly optimistic event [6,7,8,13,15,16]. We might question whether the possible preservation of periodontal cells inside the alveolus might have contributed to the clinical success of this case. However, since a minor ankylosis was noticed, this preservation is debatable and can only be confirmed via histological analysis. With regard to the periodontal ligament in late reimplantation, the role of different inflammatory mediators in periodontitis may be raised, including the formation of new mediators. Thus, new studies may be conducted in this direction, which may be seen as a new and innovative approach [11,22,23].

Concerning the ankylosis after replantation in young patients, readers must bear in mind the growth phase, because the avulsed tooth may be placed in an infra-position and disturb the alveolar and facial growth [2,9,24]. In these cases, proper management should be considered, such as decoronation, autotransplantation, resin-retained bridge, space maintainer, or orthodontic procedure, and when growth is accomplished, implant is a possibility [7,8,9,14,25].

Regarding the poor esthetic as a consequence of MTA [26,27], indirect composite veneers were revealed to be a suitable solution, considering their properties of surface texture, occlusal stability, tooth integrity, and marginal discoloration resistance [28,29].

## 4. Conclusions and Clinical Implications

The present case demonstrates that an avulsed central incisor showed satisfactory clinical outcomes after delayed replantation at four years of follow-up. The present case adds to the literature a rare occurrence of success in a case with reserved prognosis. For this reason, the IADT guidelines should be always followed and, even in extreme situations, replantation should always be considered, even if it is a temporary treatment in order to allow the most appropriate plan for each case.

## Figures and Tables

**Figure 1 medicina-56-00503-f001:**
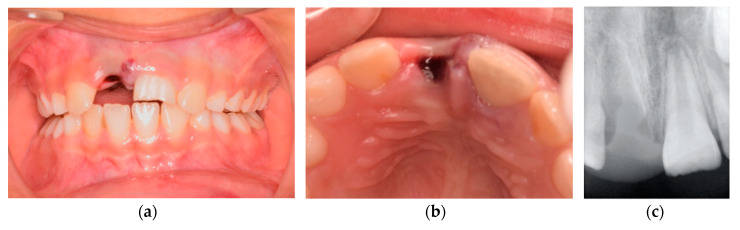
Intraoral pretreatment photographs and radiographs illustrating preoperative (**a**) buccal clinical view of anterior teeth, (**b**) occlusal clinical view of anterior teeth, and (**c**) periapical X-ray.

**Figure 2 medicina-56-00503-f002:**
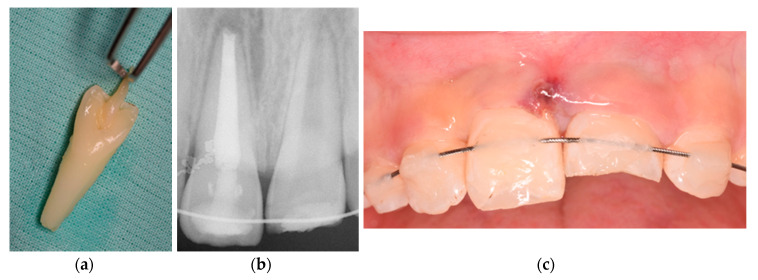
Photographs and radiographs illustrating (**a**) the avulsed tooth, (**b**) periapical X-ray after splint, and (**c**) clinical buccal view.

**Figure 3 medicina-56-00503-f003:**
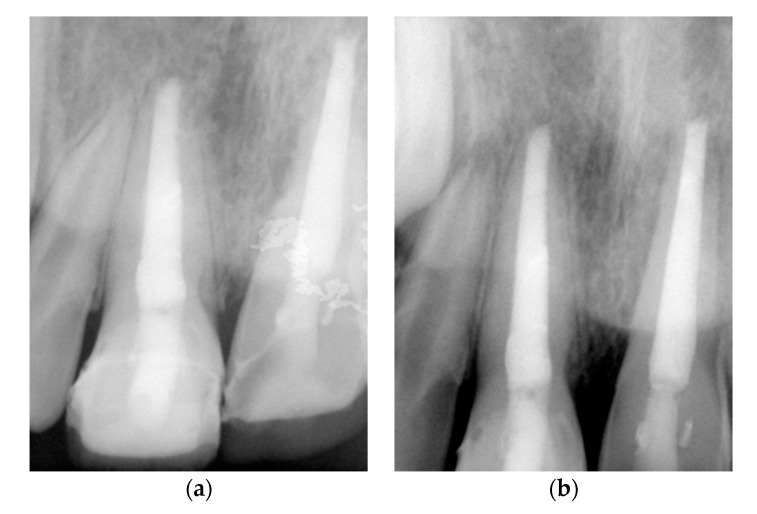
Intraoral radiographs of four years of follow-up with (**a**) a more occlusal perspective of the coronary restorations and (**b**) periapical perspective.

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
