# Peer review of "Severe Case of Delayed Replantation of Avulsed Permanent Central Incisor: A Case Report with Four-Year Follow-Up"

_medicina, 2020, doi:10.3390/medicina56100503_

Round 1

Reviewer 1 Report

The paper presents a case of delayed tooth replantation with a successful follow-up of 4 years. Even if it is a single case, its publication may be useful for clinicians in the management of similar conditions. The procedure is clearly described, but some corrections are needed.

Abstract

Line 14: please replace injury with injuries.

Line 15-16: please put “year” in plural. i.e: “years”

Line 18: the sentence “Usually complex and somewhat hopeless cases, the present case adds to the literature a rare occurrence of success. “ makes no sense. Please reword.

Case presentation

Figure 1 legend: it is not necessary to specify the patient's age here. Just describe the pictures.

Line 60:  The sentence is not well structured. Please reword. “After determining the diagnosis, and even in such extreme characteristics, delayed replantation of 1.1 was suggested, given the possible functional and esthetic consequences to the patient, and would allow to preserve bone volume for future rehabilitation with implants.”

Line 62: The sentence “For this reason, we performed the following steps as follows: 1) extra-oral pulpectomy; 2) replantation; 3) Splint with flexible orthodontic wire and composite.”  is written in the first person, i.e: “we performed”. It is desirable to follow a single way of writing. So, if you choose the passive form of verbs, please keep it throughout the manuscript. In this case: “For this reason, the following steps were performed…..”

Line 65: Did you use some solution for tooth cleaning?

Line 67: please edit “cleant” with “cleaned”

Figure 2.” Intra-oral intra-treatment radiographs of an eight-year-old girl illustrating (a) buccal view after replantation with placement of flexible orthodontic wire, (b) periapical x-ray after splint and (c) clinical buccal view. “ There is some mistake in the legend, because in  (a) picture there is not buccal view after replantation, but the avulsed tooth. Please check.

Line 81: “We, then, splinted with a flexible orthodontic wire and flowable composite (Figure 4). Amoxicillin was prescribed (every 12 hours, for 8 days), plus mouthrinse with chlorhexidine gluconate (0.2%). We also confirmed if the tetanus vaccine was up to date. Soft diet was advised for 2 weeks and a careful brushing with a soft toothbrush. “ please change in passive form. i.e: “then, the teeth were splinted with a flexible…..”

Line 87: “Pulp necrosis was diagnosed in 2.1 and we carried out apexification with MTA in a single appointment. “ please change in: “…..apexification with MTA was carried out in a single appointment.”

Line 89: at 4-years.

Please check the legend of Figure 3. “Intra-oral intra-treatment radiographs of an eight-year-old girl illustrating (a) buccal view after replantation with placement of flexible orthodontic wire, (b) periapical x-ray after splint and (c) clinical buccal view. “ is not descriptive of the pictures.

Conclusions

Line 120: the sentence “Usually complex and somewhat hopeless cases, the present case adds to the literature a rare occurrence of success.”  Is not clear. Please reword.

Author Response

We are pleased with the opportunity to revise and resubmit our manuscript now named “Severe case of delayed replantation of avulsed permanent central incisor: a case report with 4-year follow-up” (Manuscript ID medicina-930800).

Considering the editor and reviewers’ comments, all have been considered very important and were taken into profound consideration.

Manuscript changes are highlighted in the revised manuscript. Our point-by-point responses to all comments are outlined and detailed below. We hope that you find our responses satisfying.

We hope the revised manuscript will enable its further consideration. We are happy to consider further revisions and we thank you for your continued interest in our research.

Reviewer 1

The paper presents a case of delayed tooth replantation with a successful follow-up of 4 years. Even if it is a single case, its publication may be useful for clinicians in the management of similar conditions. The procedure is clearly described, but some corrections are needed.

Abstract

Line 14: please replace injury with injuries.

Line 15-16: please put “year” in plural. i.e: “years”

Line 18: the sentence “Usually complex and somewhat hopeless cases, the present case adds to the literature a rare occurrence of success. “ makes no sense. Please reword.

Answer: We replaced “injury” with “injuries”, correct “year” to “years” and rephrased that sentence to “The present case adds to the literature a rare occurrence of success in a severe case with poor prognosis.”

Case presentation

Figure 1 legend: it is not necessary to specify the patient's age here. Just describe the pictures.

Answer: We have removed the patient’s age, accordingly.

Line 60:  The sentence is not well structured. Please reword. “After determining the diagnosis, and even in such extreme characteristics, delayed replantation of 1.1 was suggested, given the possible functional and esthetic consequences to the patient, and would allow to preserve bone volume for future rehabilitation with implants.”

Answer: We rephrased to: “After determining the diagnosis, delayed replantation of 1.1 was suggested. This option was suggested given the possible functional and esthetic consequences to the patient with this tooth missing. Also, with the tooth present, bone volume would be preserved for future rehabilitation with dental implants.”

Line 62: The sentence “For this reason, we performed the following steps as follows: 1) extra-oral pulpectomy; 2) replantation; 3) Splint with flexible orthodontic wire and composite.”  is written in the first person, i.e: “we performed”. It is desirable to follow a single way of writing. So, if you choose the passive form of verbs, please keep it throughout the manuscript. In this case: “For this reason, the following steps were performed…..”

Answer: we appreciate this important remark. We followed your suggestion accordingly.

Line 65: Did you use some solution for tooth cleaning?

Answer: We have not used any solution for tooth cleaning.

Line 67: please edit “cleant” with “cleaned”

Answer: we rephrased “cleant” with “cleaned”.

Figure 2.” Intra-oral intra-treatment radiographs of an eight-year-old girl illustrating (a) buccal view after replantation with placement of flexible orthodontic wire, (b) periapical x-ray after splint and (c) clinical buccal view. “ There is some mistake in the legend, because in  (a) picture there is not buccal view after replantation, but the avulsed tooth. Please check.

Answer: We apologize for this mistake. We corrected the legend to “Photographs and radiographs illustrating (a) the avulsed tooth, (b) periapical x-ray after splint and (c) clinical buccal view.”

Line 81: “We, then, splinted with a flexible orthodontic wire and flowable composite (Figure 4). Amoxicillin was prescribed (every 12 hours, for 8 days), plus mouthrinse with chlorhexidine gluconate (0.2%). We also confirmed if the tetanus vaccine was up to date. Soft diet was advised for 2 weeks and a careful brushing with a soft toothbrush. “ please change in passive form. i.e: “then, the teeth were splinted with a flexible…..”

Answer: We have changed accordingly.

Line 87: “Pulp necrosis was diagnosed in 2.1 and we carried out apexification with MTA in a single appointment. “ please change in: “…..apexification with MTA was carried out in a single appointment.”

Answer: We have changed as suggested.

Line 89: at 4-years.

Please check the legend of Figure 3. “Intra-oral intra-treatment radiographs of an eight-year-old girl illustrating (a) buccal view after replantation with placement of flexible orthodontic wire, (b) periapical x-ray after splint and (c) clinical buccal view. “ is not descriptive of the pictures.

Answer: We sorry for this typo. We have rephrased to “Intra-oral radiographs of 4 years follow-up with (a) a more occlusal perspective of the coronary restorations and (b) periapical perspective.”

Conclusions

Line 120: the sentence “Usually complex and somewhat hopeless cases, the present case adds to the literature a rare occurrence of success.”  Is not clear. Please reword.

Answer: We reworded to “The present case adds to the literature a rare occurrence of success in a case with reserved prognosis.”

Reviewer 2 Report

This paper, which is a case report.

The periapical RX are not the best one, although there are taken in an "easy" clinical region...

Please find hereunder some questions :

  • what do you mean with "a rare occurence of success" with this case ?
  • the postoperative sucess after 4 years is "a short term " one in the field of dental trauma. The patient will then be only 12 years old !
  • not any postoperative clinical view is shown : why ?
  • in the discussion, which is very short, not any mention is done about the vitality (preservation) of the (maybe) persistant periodontal cells in the alveolus, which may be the key of the success !
  • The legends of the figures 1, 2a and 3 are not correct or don't correspond to the pictures ! Fig 3 has no "c" !
  • L 63 : splint instead of Splint
  • L 68 : ethylene instead of Ethylene
  • L 85 : the authors mention the pulp status, but of which teeth ?

Author Response

We are pleased with the opportunity to revise and resubmit our manuscript now named “Severe case of delayed replantation of avulsed permanent central incisor: a case report with 4-year follow-up” (Manuscript ID medicina-930800).

Considering the editor and reviewers’ comments, all have been considered very important and were taken into profound consideration.

Manuscript changes are highlighted in the revised manuscript. Our point-by-point responses to all comments are outlined and detailed below. We hope that you find our responses satisfying.

We hope the revised manuscript will enable its further consideration. We are happy to consider further revisions and we thank you for your continued interest in our research.

Reviewer 2

This paper, which is a case report.

The periapical RX are not the best one, although there are taken in an "easy" clinical region...

Answer: We acknowledge that these X-rays are not perfect, but we believe they are supported good quality photographs.

Please find hereunder some questions :

  • what do you mean with "a rare occurence of success" with this case ?

Answer: We used "a rare occurence of success" because this cases normally present severe and dramatic root resorption within the first 1-2 years of follow-up.

  • the postoperative sucess after 4 years is "a short term " one in the field of dental trauma. The patient will then be only 12 years old !

Answer: In fact, 4 years is usually considered a short-term follow-up. However, it is considered a minimum acceptable follow-up. However we intend to follow this case and present longer follow-ups in the future.

  • not any postoperative clinical view is shown : why ?

Answer: We added one postoperative clinical view, Figure 2C. We decided to not present any other photograph because our main focus was the success of the replantation procedure and the absence of negative events in the subsequent follow-ups.

  • in the discussion, which is very short, not any mention is done about the vitality (preservation) of the (maybe) persistant periodontal cells in the alveolus, which may be the key of the success!

Answer: We have not mentioned the possible vitality (preservation) of the persistent periodontal cells in the alveolus because we reported minor signs of ankylosis, which may contradict this supposition. Nevertheless, we added the sentence “We might question whether the possible preservation of periodontal cells inside the alveolus might contributed to clinical success of this case, however since minor ankylosis was noticed this preservation is debatable and only possible to confirm via histological analysis.” (Lines 106-109), to refer this important remark.

  • The legends of the figures 1, 2a and 3 are not correct or don't correspond to the pictures ! Fig 3 has no "c" !

Answer: We apologize for these mistakes. We corrected them accordingly.

  • L 63 : splint instead of Splint

Answer: we have corrected accordingly.

  • L 68 : ethylene instead of Ethylene
  • Answer: we have reworded it accordingly.
  •  
  • L 85 : the authors mention the pulp status, but of which teeth ?

Answer: we added “in both teeth”. We appreciate it.

Reviewer 3 Report

The English language must be improved.

The paper present some errors in the text:

Line 19 ?? text not clear

line 42 her parents

line 49 in dry condition without parenthesis

line  51 fracture of upper left central incisor (2.1)

line 52 thermal test

line 58 (b) occlusal clinical view

line 59 (c) the image is not correct  and  needs to be turned around

line 66 with MTA for apecification?

line 74 figure 2: c is a and a is c (they have been switched around)

line 80 replanted,

line 81 radiographically confirmed and the splinted……..

line 82 figure 4 is not present

line 85 pulp status of 2.1

line 95 figure 3: the caption does not match the images a and b

What type of treatment do you reserve to the periodontal ligament? (line 63, line 65)

Author Response

We are pleased with the opportunity to revise and resubmit our manuscript now named “Severe case of delayed replantation of avulsed permanent central incisor: a case report with 4-year follow-up” (Manuscript ID medicina-930800).

Considering the editor and reviewers’ comments, all have been considered very important and were taken into profound consideration.

Manuscript changes are highlighted in the revised manuscript. Our point-by-point responses to all comments are outlined and detailed below. We hope that you find our responses satisfying.

We hope the revised manuscript will enable its further consideration. We are happy to consider further revisions and we thank you for your continued interest in our research.

Reviewer 3

The English language must be improved.

The paper present some errors in the text:

Line 19 ?? text not clear

Answer: We have corrected to “The present case adds to the literature a rare occurrence of success in a severe case with poor prognosis.”.

line 42 her parents

Answer: We have corrected it.

line 49 in dry condition without parenthesis

Answer: we removed the parenthesis

line  51 fracture of upper left central incisor (2.1)

Answer: we added it.

line 52 thermal test

Answer: we rephrased it.

line 58 (b) occlusal clinical view

Answer: we changed to “occlusal”.

line 59 (c) the image is not correct  and  needs to be turned around

Answer: We appreciate this remark. We have turned the image.

line 66 with MTA for apecification?

Answer: we added “MTA placed for apexification”.

line 74 figure 2: c is a and a is c (they have been switched around)

Answer: We corrected the legend accordingly.

line 80 replanted,

Answer: we added it.

line 81 radiographically confirmed and the splinted……..

Answer: We rephrased to “radiographically confirmed and, then, splinted with a flexible orthodontic (...)”.

line 82 figure 4 is not present

Answer: We have removed it. This was a typo.

line 85 pulp status of 2.1

Answer: We added it accordingly.

line 95 figure 3: the caption does not match the images a and b

Answer: We have corrected it to “Figure 3. Intra-oral radiographs of 4 years of follow-up with (a) a more occlusal perspective of the coronary restorations and (b) periapical perspective.”

What type of treatment do you reserve to the periodontal ligament? (line 63, line 65).

Answer: We have tried to maintain the root surface as best cleaned as possible as described. Also, we have written previously “the tooth was kept in a 2% sodium fluoride (NaF) solution for 20 min, because of the possibility to slow down osseous replacement”.

Round 2

Reviewer 2 Report

Although some responses by the authors to the reviewer's comments or questions were appropriate, I as actual reviewer did not intend to change my mind.

I believe strongly this case report doesn't add anything nor any value to the scientific literature on the subject.

Author Response

Nothing to declare. We answered to the Academic Editor fine recommendations